# YM155-Adapted Cancer Cell Lines Reveal Drug-Induced Heterogeneity and Enable the Identification of Biomarker Candidates for the Acquired Resistance Setting

**DOI:** 10.3390/cancers12051080

**Published:** 2020-04-26

**Authors:** Martin Michaelis, Mark N. Wass, Ian Reddin, Yvonne Voges, Florian Rothweiler, Stephanie Hehlgans, Jaroslav Cinatl, Marco Mernberger, Andrea Nist, Thorsten Stiewe, Franz Rödel, Jindrich Cinatl

**Affiliations:** 1Industrial Biotechnology Centre and School of Biosciences, University of Kent, Canterbury CT2 7NJ, UK; M.Michaelis@kent.ac.uk (M.M.); M.N.Wass@kent.ac.uk (M.N.W.); ir84@kent.ac.uk (I.R.); 2Institute for Medical Virology, University Hospital, Goethe University Frankfurt am Main, 60596 Frankfurt, Germany; yvonnevoges@gmx.de (Y.V.); f.rothweiler@kinderkrebsstiftung-frankfurt.de (F.R.); ja.cinatl@kinderkrebsstiftung-frankfurt.de (J.C.); 3Department of Radiotherapy and Oncology, University Hospital, Goethe University Frankfurt am Main, 60590 Frankfurt, Germany; Stephanie.Hehlgans@kgu.de (S.H.); Franz.Roedel@kgu.de (F.R.); 4Institute of Molecular Oncology, Member of the German Center for Lung Research (DZL), Philipps-University, 35032 Marburg, Germany; marco.mernberger@imt.uni-marburg.de (M.M.); stiewe@uni-marburg.de (T.S.); 5Genomics Core Facility, Philipps-University, 35043 Marburg, Germany; andrea.nist@imt.uni-marburg.de

**Keywords:** acquired drug resistance, biomarkers, therapy monitoring, neuroblastoma, BIRC5, survivin, intrinsic drug resistance

## Abstract

Survivin is a drug target and its suppressant YM155 a drug candidate mainly investigated for high-risk neuroblastoma. Findings from one YM155-adapted subline of the neuroblastoma cell line UKF-NB-3 had suggested that increased ABCB1 (mediates YM155 efflux) levels, decreased SLC35F2 (mediates YM155 uptake) levels, decreased survivin levels, and *TP53* mutations indicate YM155 resistance. Here, the investigation of 10 additional YM155-adapted UKF-NB-3 sublines only confirmed the roles of ABCB1 and SLC35F2. However, cellular ABCB1 and SLC35F2 levels did not indicate YM155 sensitivity in YM155-naïve cells, as indicated by drug response data derived from the Cancer Therapeutics Response Portal (CTRP) and the Genomics of Drug Sensitivity in Cancer (GDSC) databases. Moreover, the resistant sublines were characterized by a remarkable heterogeneity. Only seven sublines developed on-target resistance as indicated by resistance to RNAi-mediated survivin depletion. The sublines also varied in their response to other anti-cancer drugs. In conclusion, cancer cell populations of limited intrinsic heterogeneity can develop various resistance phenotypes in response to treatment. Therefore, individualized therapies will require monitoring of cancer cell evolution in response to treatment. Moreover, biomarkers can indicate resistance formation in the acquired resistance setting, even when they are not predictive in the intrinsic resistance setting.

## 1. Introduction

Sepantronium bromide (YM155) was introduced as an anti-cancer drug candidate that inhibits expression of the *BIRC5* gene, which encodes the protein survivin [1]. In the meantime, YM155 has been suggested to exert additional and/or alternative mechanisms of anticancer actions, including induction of DNA damage, inhibition of NFκB signaling, induction of death receptor 5 expression, and/or suppression of MCL-1, XIAP, cIAP-1/2, BCL-2, BCL-XL, FLIP, EGFR, and/or mTORC [2,3,4,5,6,7,8,9,10,11,12,13].

A number of studies have investigated the potential of YM155 against neuroblastoma cells [14,15,16,17]. Neuroblastoma is the most common extracranial solid childhood tumor. Treatment outcomes in high-risk neuroblastoma patients remain unsatisfactory. About 50% of these patients relapse and have a 5-year-survial rate below 10% [18,19,20,21]. We have recently shown that suppression of survivin expression is the main mechanism through which YM155 exerts its anti-neuroblastoma effects [16]. Notably, the New Drug Development Strategy (NDDS, a project of Innovative Therapies for Children with Cancer, the European Network for Cancer Research in Children and Adolescents, and the International Society of Paediatric Oncology Europe Neuroblastoma) has categorized survivin as a high priority drug target in neuroblastoma and YM155 as a high priority drug [22].

The formation of acquired resistance is a central problem in (metastasized) cancer diseases that need to be treated by systemic drug therapy. Although many cancers initially respond well to therapy, resistance formation is common, and cures are rare [23]. Hence, biomarkers that indicate early therapy failure are needed to adapt therapies if resistance emerges. Liquid biopsies (e.g., circulating tumor cells) enable the monitoring of cancer cell evolution in patients with ever more detail [24]. However, the translation of the resulting information into improved therapies is hampered by a lack of understanding of the processes underlying acquired resistance formation and, in turn, a lack of biomarkers.

Most studies focus on biomarkers that indicate whether a certain cancer cell (population) is likely to respond to a certain treatment but not on biomarkers that indicate early that a current therapy has stopped working. This also applies to the previous studies that investigated the efficacy of YM155 in neuroblastoma [14,15,17]. However, it is known that intrinsic and acquired resistance mechanisms may substantially differ [25,26,27]. Using a single YM155-adapted neuroblastoma cell line, we identified increased ABCB1 (also known as P-glycoprotein or MDR1) expression, decreased SLC35F2 (solute carrier family 35 member F2) expression, decreased survivin expression, and loss-of-p53 function as potential markers of resistance formation to YM155 [16]. Given the tremendous (intra-tumor) heterogeneity in cancer [28], it is likely that the processes, which result in acquired resistance formation, are equally complex. If so, then a larger number of models of acquired resistance to a certain drug will be needed to adequately address the complexity of the resistance formation process.

To test this hypothesis, we here established and characterized 10 further YM155-adapted UKF-NB-3 neuroblastoma cell lines. Moreover, acquired resistance models may provide information that cannot be gained from the comparison of non-adapted cell lines with a varying resistance status. To investigate whether this is the case, the findings from the YM155-adapted UKF-NB-3 sublines were compared to data from the two large pharmacogenomics screens Genomics of Drug Sensitivity in Cancer (GDSC) and Cancer Therapeutic Response Portal (CTRP), which use non-adapted cancer cell lines [29,30], whether we can obtain information from our acquired resistance models that cannot be identified from traditional approaches using non-adapted cell lines. We also analyzed YM155 response data from the two large pharmacogenomics screens Genomics of Drug Sensitivity in Cancer (GDSC) and Cancer Therapeutic Response Portal (CTRP) [29,30]. We found a remarkable heterogeneity between the individual sublines, although they all were derived from the same parental cell line. An increase in cellular ABCB1 levels and/or a decrease in SLC35F2 levels indicate resistance formation to YM155, although the ABCB1 and/or SLC35F2 levels cannot be used to infer YM155 sensitivity in YM155-naïve cell lines. The use of the panel of YM155-adapted cell lines further enabled us to show that the cellular survivin levels and the *TP53* status do not reliably indicate resistance formation.

## 2. Results

### 2.1. YM155-Adapted UKF-NB-3 Sublines Display Pronounced YM155 Resistance

All YM155-adapted UKF-NB-3 sublines displayed pronounced YM155 resistance (Figure 1, Appendix A). The YM155-adapted UKF-NB-3 sublines displayed between 38- and 76-fold increased YM155 IC50 values and between 30- and 135-fold increased IC90 values relative to UKF-NB-3 (Appendix A). Representative photos of the morphology of the project cell lines are presented in Appendix A and the doubling times in Appendix A.

### 2.2. The Cellular TP53 Status is Not a Reliable Indicator of YM155 Sensitivity

Originally, the cellular *TP53* status was described to not directly influence the anticancer action of YM155 [31]. In agreement, the analysis of the Genomics of Drug Sensitivity in Cancer (GDSC) and Cancer Therapeutics Response Portal (CTRP) databases did not indicate differences in the YM155 sensitivity between cell lines in dependence on their *TP53* status (wild-type or mutant) (Figure 2).

However, the activation of p53 signaling seems to be involved in the anticancer mechanism of action of YM155 at least in some cancer cells. We have previously shown in neuroblastoma cells that YM155 activates p53 signaling, that p53 activation using MDM2 inhibitors enhances the YM155 effects, and that p53 depletion reduces cancer cell sensitivity to YM155 [15]. In addition, a YM155-adapted UKF-NB-3 subline harbored a *TP53* mutation [16]. However, all 10 YM155-adapted UKF-NB-3 sublines that we investigated here displayed wild-type *TP53* as indicated by *TP53* next-generation sequencing. The cellular p53 levels also did not differ consistently between UKF-NB-3 and its YM155-resistant sublines (Appendix A). The YM155-resistant UKF-NB-3 sublines remained similarly sensitive to the MDM2 inhibitor and p53 activator nutlin-3 as UKF-NB-3 (Appendix A). Hence, our findings do not suggest that YM155 adaption is generally associated with a loss of p53 function in neuroblastoma cells. The cellular *TP53* status is not a reliable indicator of YM155 sensitivity, neither in the intrinsic nor in the acquired resistance setting.

### 2.3. Cellular Survivin Levels Do Not Reliably Indicate YM155 Response

Some studies suggested cancer cells with high survivin levels to be particularly sensitive to YM155 [31,32,33]. However, other studies failed to detect an association between the cellular survivin status and YM155 activity [16,34]. When we compared the YM155 sensitivity between cancer cell lines with high and low survivin expression, we found statistically significant differences across all cell lines in the GDSC and CTRP datasets but not across the neuroblastoma cell lines (Figure 3). It was not possible to predict whether a certain cell line was sensitive to YM155 based on the cellular survivin level (Figure 3). 

Notably, a YM155-adapted UKF-NB-3 subline had previously displayed reduced survivin levels relative to the parental cell line [16]. However, the analysis of the 10 additional YM155-adapted UKF-NB-3 sublines in this study revealed that resistance acquisition to YM155 was not associated with a consistent change in the survivin expression patterns (Appendix A).

### 2.4. Acquired YM155 Resistance is Associated with Decreased Sensitivity to Survivin Depletion

Our previous findings suggested that YM155 predominantly exerts its antineuroblastoma effects via suppression of survivin expression [16]. Seven of the 10 YM155-adapted UKF-NB-3 sublines (I, III, V, VII, VIII, IX, X) displayed decreased sensitivity to siRNA-mediated survivin depletion, indicating that they developed on-target resistance. However, two sublines were similarly sensitive as parental UKF-NB-3 cells (II, VI), and one subline (IV) was more sensitive to survivin depletion (Figure 4, Appendix A). This shows that the YM155 resistance mechanisms differ between the individual UKF-NB-3 sublines. Notably, the viability of all sublines is still affected by survivin depletion, which shows that they have retained some level of survivin dependence.

### 2.5. Relevance of Cellular ABCB1 and SLC35F2 Levels in the Context of YM155 Resistance

Increased cellular ABCB1 (mediates YM155 efflux) levels and decreased SLC35F2 (mediates cellular YM155 uptake) levels have previously been identified as important YM155 resistance mechanisms [14,16,17,35]. To further investigate the relationship between ABCB1 and SLC35F2 levels and YM155 sensitivity, we compared the YM155 sensitivity in cell lines that displayed low or high expression of the respective genes using GDSC and CTRP data. In agreement with previous data, high *ABCB1* expression (Figure 5) and low *SLC35F2* expression (Figure 6) were associated with reduced YM155 sensitivity. When we used transcriptomics data from the GDSC and CTRP to correlate the expression of all genes with YM155 sensitivity, *ABCB1* ranked as the gene whose expression was most strongly correlated to the YM155 AUC (area under the curve, unit used to quantify the drug response) (Table 1) in the GDSC and CTRP. *SLC35F2* expression was most strongly inversely correlated to the YM155 AUC (Table 2) in both data sets. There were no further overlaps among the top 10 genes between the two databases (Table 1 and Table 2). However, the YM155 sensitivity of a certain cell line could not be reliably predicted based on the cellular ABCB1 and/or SLC35F2 levels (Figure 5 and Figure 6).

All YM155-adapted UKF-NB-3 sublines displayed increased ABCB1 levels relative to UKF-NB-3 (Figure 7, Appendix A). Acquired YM155 resistance was also generally associated with decreased SLC35F2 levels, in particular in the sublines I, IV, VI, and X (Figure 7, Appendix A). This indicates that increased ABCB1 levels and decreased SLC35F2 levels have potential as biomarkers, indicating YM155 resistance formation in response to YM155-based therapies, although cellular ABCB1 and SLC35F2 levels do not enable the prediction of YM155 sensitivity in YM155-naïve cells.

### 2.6. YM155-Adapted UKF-NB-3 Cells Remain Sensitive to DNA Damage Caused by Irradiation and Cytotoxic Drugs

YM155 has been proposed to exert its anticancer effects via the induction of DNA damage in some experimental systems [3,5,35,36]. To study whether the acquisition of YM155 resistance was associated with a generally increased resistance to DNA damage, UKF-NB-3 and its YM155-adapted UKF-NB-3 sublines were irradiated at a dose range of one to five Gy. None of the YM155-adapted UKF-NB-3 sublines displayed substantially reduced sensitivity to irradiation relative to UKF-NB-3 (Figure 8). Moreover, none of the YM155-resistant UKF-NB-3 sublines displayed reduced sensitivity to cisplatin (causes DNA crosslinks) or topotecan (topoisomerase I inhibitor), which cause DNA damage by different mechanisms (Figure 8, Appendix A). There was also no coherent increase in resistance to the nucleoside analogue gemcitabine (Figure 8, Appendix A). These data do not suggest that the activity of YM155 against UKF-NB-3 cells would predominantly depend on DNA damage induction.

### 2.7. Heterogeneity among YM155-Adapted UKF-NB-3 Sublines

While the YM155-adapted UKF-NB-3 sublines displayed limited heterogeneity in response to treatment with cisplatin and topotecan, remarkable differences in the gemcitabine IC_50_s were detected (Figure 8, Appendix A). The fold difference between the YM155-adapted subline with the lowest gemcitabine IC_50_ (V, 0.12 ng/mL) and the subline with the highest IC_50_ (VIII, 0.65 ng/mL) was 5.4-fold. This heterogeneity is in agreement with the up to 29-fold difference observed in the cell viability in response to BIRC5/survivin depletion between the most sensitive (IV) and the most resistant (VII) subline (Figure 4). Resistance profiles to the destabilizing tubulin-binding agent vincristine also revealed a substantial heterogeneity between the YM155-resistant UKF-NB-3 sublines (Figure 9, Appendix A), resulting in a fold difference of 127 between subline VI (vincristine IC_50_: 714 ng/mL) and subline IX (vincristine IC_50_: 5.6 ng/mL) (Appendix A).

## 3. Discussion

In a previous study, a YM155-adapted subline of the neuroblastoma cell line UKF-NB-3 was characterized by increased cellular ABCB1 levels, decreased SLC35F2 and survivin levels, and a *TP53* mutation [16]. Here, we systematically investigated the relevance of cellular ABCB1, SLC35F2, and survivin levels as well as the *TP53* status as potential biomarkers of YM155 resistance formation in the intrinsic resistance setting, using data derived from the GDSC and CTRP databases [29,30], and in the acquired resistance setting, using an additional set of 10 YM155-adapted UKF-NB-3 sublines, which were established in independent experiments.

Increased ABCB1 expression (mediates YM155 efflux) and decreased SLC35F2 expression (mediates cellular YM155 uptake) were identified as YM155 resistance mechanisms in panels of YM155-naïve cell lines that displayed varying levels of these proteins and in functional studies [14,16,17,37], which was further supported by our analysis of GDSC and CTRP data [29,30]. Despite their roles in determining YM155 resistance, however, cellular ABCB1 or SLC35F2 levels did not enable the prediction of whether an individual cell line would be sensitive to YM155 or not. The YM155-adapted UKF-NB-3 cell lines generally displayed elevated cellular ABCB1 levels and reduced SLC35F2 levels relative to UKF-NB-3. Hence, an increase in the cellular ABCB1 levels and/or a decrease in the SLC35F2 levels have potential as biomarkers that indicate resistance formation, even though the respective cellular levels do not reliably predict the YM155 response in YM155-naïve cells.

Initially, the *TP53* status was reported not to influence the anticancer effects of YM155 [31], which was further supported by our analysis of GDSC and CTRP data. In neuroblastoma cells, however, YM155 induced p53 signaling, p53 depletion reduced YM155 sensitivity, and a YM155-adapted UKF-NB-3 subline harbored a *TP53* mutation [16]. Here, all 10 YM155-adapted UKF-NB-3 sublines retained wild-type *TP53*. Thus, the role of p53 seems to depend on the individual cellular context. Neither the cellular *TP53* status nor the formation of *TP53* mutations can currently be considered as valid biomarkers for YM155 therapies.

The relevance of cellular survivin levels for cancer cell sensitivity to YM155 is not clear [16,31,32,33,34,35]. Our analysis of GDSC and CTRP data indicated that high survivin (BIRC5) expression was associated with increased YM155 sensitivity. However, it was not possible to infer the YM155 sensitivity of a particular cell line based on its survivin status. Reasons for this may include that survivin is not in all cell lines the major therapeutic target of YM155 as it is in neuroblastoma cells [1,2,3,4,5,6,7,8,9,10,11,12,16,36] and/or that off-target resistance mechanisms, such as ABCB1 and SLC35F2 expression, may affect YM155 efficacy independently of the survivin status [14,16,17,35].

The YM155-adapted UKF-NB-3 sublines displayed various survivin levels, demonstrating that resistance formation to YM155 is also not associated with a consistent change in cellular survivin levels. Seven of the YM155-adapted cell lines displayed on-target resistance as indicated by reduced sensitivity to RNAi-mediated BIRC5/survivin depletion relative to parental UKF-NB-3 cells, further confirming that survivin is a target of YM155 in neuroblastoma cells. However, cellular survivin levels do not represent a reliable biomarker of resistance formation to YM155.

While YM155 was described to act via the induction of DNA damage in some cancer types [3,5,35,36], our previous results did not indicate a causative role of DNA damage induction in the anticancer effects of YM155 against neuroblastoma cells [16]. YM155 resistance formation in the YM155-adapted neuroblastoma cell lines was also not associated with generally decreased sensitivity to radiation or DNA damage caused by cisplatin (causes DNA crosslinks), gemcitabine (nucleoside analogue), or topotecan (topoisomerase I inhibitor). This indicates that YM155 resistance formation in neuroblastoma cells is not generally associated with an increased resistance to DNA damage induction.

In this study, the use of multiple models of acquired resistance enabled insights that could not be gained from just one drug-adapted subline. The previous investigation of one YM155-resistant UKF-NB-3 subline had suggested that changes in the cellular *TP53* status and survivin levels indicate resistance formation [16], which was not confirmed in our current panel of 10 YM155-adapted UKF-NB-3 sublines. Moreover, the use of multiple sublines provided a novel glimpse into the significant heterogeneity of the resistance formation process, even though all resistant sublines were derived from the same parental cell line. Only 7 of the 10 sublines developed on-target resistance mechanisms as indicated by reduced sensitivity to survivin depletion. The sublines also showed substantial variation in their sensitivity to irradiation (up to 7-fold difference at 5Gy), gemcitabine (up to 5-fold), and vincristine (up to 127-fold). Notably, a much higher heterogeneity would be expected in the clinical situation, in which tumors are already characterized by much higher heterogeneity than cancer cell lines and in which combination therapies are common.

## 4. Materials and Methods

### 4.1. Cells

The MYCN-amplified neuroblastoma cell line UKF-NB-3 was established from a bone marrow metastasis of a stage IV neuroblastoma patient [38]. Ten YM155-resistant UKF-NB-3 sublines were derived from the resistant cancer cell line (RCCL) collection (https://research.kent.ac.uk/industrial-biotechnology-centre/the-resistant-cancer-cell-line-rccl-collection/). They were established by adaptation of UKF-NB-3 cells (passage 87) to growth in the presence of YM155 20nM by previously described methods [39] and designated as UKF-NB-3^r^YM155^20nM^I to UKF-NB-3^r^YM155^20nM^X. All cells were propagated in IMDM supplemented with 10 % FBS, 100 IU/mL penicillin, and 100 µg/mL streptomycin at 37 °C. Cells were routinely tested for mycoplasma contamination (PlasmoTest™, Mycoplasma Detection kit, InvivoGen, Toulouse, France) and authenticated by short tandem repeat profiling.

To determine doubling times, 2 × 10^4^ cells per well were plated into 6-well plates, incubated at 37 °C and 5% CO_2_, and counted after 1, 2, 3, 5, and 7 days using a Neubauer chamber. Doubling times were then calculated using http://www.doubling-time.com/compute.php, which uses the equation:

Doubling time = duration × log(2)/log(final cell number) − log(initial cell number).

### 4.2. Viability Assay

Cell viability was tested by the 3-(4,5-dimethylthiazol-2-yl)-2,5-diphenyltetrazolium bromide (MTT) dye reduction assay after 120 h of incubation modified as described previously [39]. Cells (2 × 10^4^/100 µL per well in 96-well plates) were incubated in the presence or absence of drug for 120 h. Then, 25 µL of MTT solution (2 mg/mL (*w/v*) in PBS) were added per well, and the plates were incubated at 37 °C for an additional 4 h. After this, the cells were lysed using 100 µL of a buffer containing 20% (*w/v*) sodium dodecylsulfate and 50% (*v/v*) *N,N*-dimethylformamide with the pH adjusted to 4.7 at 37 °C for 4 h. Absorbance was determined at 570 nm for each well using a 96-well multiscanner. After subtraction of the background absorption, the results were expressed as the percentage viability relative to control cultures that received no drug. Drug concentrations that inhibited cell viability by 50% (IC50) or 90% (IC90) were determined using CalcuSyn (Biosoft, Cambridge, UK).

### 4.3. TP53 Next-Generation Sequencing

*TP53* next-generation sequencing was performed as previously described [16]. All coding exonic and flanking intronic regions of the human *TP53* gene were amplified from genomic DNA with Platinum™ Taq DNA polymerase (Life Technologies) by multiplex PCR using two primer pools with 12 non-overlapping primer pairs each, yielding approximately 180 bp amplicons. Each sample was tagged with a unique 8-nucleotide barcode combination using 12 differently barcoded forward and eight differently barcoded reverse primer pools. Barcoded PCR products from up to 96 samples were pooled, purified, and an indexed sequencing library was prepared using the NEBNext^®^ ChIP-Seq Library Prep Master Mix Set for Illumina in combination with NEBNext^®^ Multiplex Oligos for Illumina (New England Biolabs). The quality of sequencing libraries was verified on a Bioanalyzer DNA High Sensitivity chip (Agilent) and quantified by digital PCR. 2 × 250 bp paired-end sequencing was carried out on an Illumina MiSeq (Illumina) according to the manufacturer’s recommendations at a mean coverage of 300×.

Read pairs were demultiplexed according to the forward and reverse primers and subsequently aligned using the Burrows-Wheeler Aligner against the Homo sapiens Ensembl reference (rev. 79). Overlapping mate pairs where combined and trimmed to the amplified region. Coverage for each amplicon was calculated via SAMtools (v1.1) [40]. To identify putative mutations, variant calling was performed using SAMtools in combination with VarScan2 (v2.3.9) [41]. Initially, SAMtools was used to create pileups with a base quality filter of 15. Duplicates, orphan reads, unmapped, and secondary reads were excluded. Subsequently, Varscan2 was applied to screen for SNVs and InDels separately, using a low-stringency setting with minimal variant frequency of 0.1, a minimum coverage of 20, and a minimum of 10 supporting reads per variant to account for cellular and clonal heterogeneity. Minimum average quality was set to 20 and a strand filter was applied to minimize miscalls due to poor sequencing quality or amplification bias. The resulting list of putative variants was compared against the IARC *TP53* (R17) database to check for known p53 cancer mutations.

### 4.4. Western Blot

Cells were lysed using Triton-X-100 sample buffer, and proteins were separated by SDS-PAGE. Detection occurred by using specific antibodies against β-actin (1:5000 dilution, Biovision through BioCat GmbH, Heidelberg Germany; secondary antibody: IRDye^®^ 800CW Goat anti-Mouse IgG, dilution 1:25,000, Li-Cor Biosciences, Lincoln, NE, USA), SLC35F2 (1:200, Santa Cruz Biotechnology, Dallas, TX, USA; secondary antibody: IRDye^®^ 800CW Goat anti-Mouse IgG, dilution 1:25,000, Li-Cor Biosciences, Lincoln, NE, USA), GAPDH (1:4000, Trevigen via Bio-Techne GmbH, Wiesbaden, Germany; secondary antibody: IRDye^®^ 800CW Goat anti-Rabbit IgG, dilution 1:25,000, Li-Cor Biosciences), ABCB1 (1:1,000, Cell Signaling via New England Biolabs, Frankfurt, Germany; secondary antibody: IRDye^®^ 800CW Goat anti-Rabbit IgG, dilution 1:25,000, Li-Cor Biosciences), p53 (1:1000, Enzo Life Sciences, Lörrach, Germany; secondary antibody: IRDye^®^ 800CW Goat anti-Mouse IgG, dilution 1:25,000, Li-Cor Biosciences), and survivin (1:500, R&D Systems, Minneapolis, MN, USA; secondary antibody: IRDye^®^ 800CW Goat anti-Rabbit IgG, dilution 1:25,000, Li-Cor Biosciences). Protein bands were visualized by laser-induced fluorescence using infrared scanner for protein quantification (Odyssey, Li-Cor Biosciences) and Image Studio Ver. 5.2 software (Li-Cor Biosciences) for densitometric analyses.

### 4.5. RNA Interference Experiments

Transient depletion of BIRC5/survivin was achieved using synthetic siRNA oligonucleotides (ON-TARGETplus SMARTpool) from Dharmacon (Lafayette, CO, USA). Non-targeting siRNA (ON-TARGETplus SMARTpool) was used as negative control. Cells were transfected by electroporation using the NEON Transfection System (Invitrogen, Darmstadt; Germany) according to the manufacturer protocol. Cells were grown to 60–80% confluence, trypsinised, and 1.2 × 10^6^ cells were re-suspended in 200 µL of resuspension buffer R including 2.5 µM siRNA. The electroporation was performed using two 20 ms pulses of 1400 V. Subsequently, the cells were transferred into cell culture plates or flasks, containing pre-warmed cell culture medium. During the set-up of the experiments, the SMARTpool was compared to two individual siRNAs (target sequences: GCAAAGGAAACCAACAAUA, GGAAAGGAGAUCAACAUUU) (Appendix A).

### 4.6. Irradiation Procedure

In 96-well cell culture plates, 10^4^ cells per well were irradiated at room temperature (Greiner, Bio-ONE GmbH, Frickenhausen, Germany) with single doses of X-rays ranging from 1 to 5 Gy using a linear accelerator (SL 75/5, Elekta, Crawley, UK) with 6 MeV photons/100 cm focus–surface distance with a dose rate of 4.0 Gy/min. Sham-irradiated cultures were kept at room temperature in the X-ray control room while the other samples were irradiated.

### 4.7. Analysis of Data Derived from Large Pharmacogenomic Studies

All data (including drug response area under curve (AUC) data for YM-155-treated cancer cell lines, basal gene-expression for ABCB1, BIRC5 (the gene that encodes survivin), and SLC35F2, and genomic alterations of p53) in this study were obtained from two online resources: Version 2 of the Cancer Therapeutics Response Portal (CTRP v2) data [29,42] were obtained from the Cancer Target Discovery and Development (CTD^2^) data portal (ocg.cancer.gov/programs/ctd2/data-portal). The Genomics of Drug Sensitivity in Cancer (GDSC) data were obtained from www.cancerrxgene.org [43,44].

The CTRP contains ABCB1, BIRC5, and SLC35F2 expression data for 823 cell lines and YM-155 AUC data for 715 cell lines. For 703 cell lines (including 12 neuroblastoma cell lines), gene expression data and YM155 AUC values were available. Whole exome sequencing (WES) data was available for 546 of the cell lines for which YM-155 sensitivity data was also available (including 11 neuroblastoma cell lines).

The GDSC contains ABCB1, BIRC5, and SLC35F2 expression data for 1019 cell lines and YM155 AUC data for 945 cell lines. Expression data and WES data were available for all 945 cell lines with YM-155 sensitivity data (including 30 neuroblastoma cell lines).

Data processing was performed using Perl version 5.26.0, and R statistical packages version 3.3.2. Cell lines were determined to display either high or low expression for each gene using the median gene expression as a threshold (i.e., low expression < = median expression, high expression > median expression). Box plots indicating the YM-155 sensitivity in cell lines that display low or high expression of a certain gene or wild-type or mutant *TP53* were produced using the ggplot2 package [45] in R.

Statistical tests were carried out in R and included Wilcoxon rank-sum test [46] and Pearson’s correlation [47]. Correction for multiple comparisons was performed using the Benjamini–Hochberg procedure [37].

### 4.8. Statistics

Results are expressed as mean ± S.D. of at least three experiments. Comparisons between the means of two sample groups were performed using Student’s *t*-test. The means of three or more sample groups were compared by one-way ANOVA followed by the Student–Newman–Keuls test. *P* values lower than 0.05 were considered to be significant.

## 5. Conclusions

Our data revealed a high phenotypic heterogeneity among a panel of 10 YM155-resistant sublines of the neuroblastoma cell line UKF-NB-3. This heterogeneity is of conceptual importance, because it shows that even a defined cancer cell population of limited intrinsic heterogeneity can develop various resistance mechanisms and phenotypes in response to treatment. From a clinical perspective, this means that the close monitoring of cancer cell evolution in response to therapy will have to become an essential part of the design of individualized therapies. Notably, such insights can only be gained from preclinical model systems, such as drug-adapted cancer cell lines, which enable the repeated adaptation of a given cancer cell population to the same treatment but not from clinical material as every patient can only be treated once.

Our findings also demonstrate that biomarkers can indicate resistance formation, even when they do not enable the prediction of drug sensitivity in therapy-naïve cancer cells. Hence, the use of biomarkers differs between the intrinsic and the acquired resistance setting, and pre-clinical models of acquired drug resistance are needed for the identification of such biomarkers that herald resistance development.

## Figures and Tables

**Figure 1 cancers-12-01080-f001:**
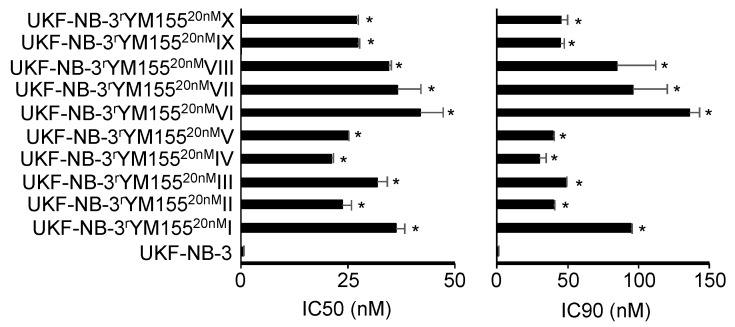
YM155 concentrations that reduce the viability of UKF-NB-3 and its YM155-adapted sublines by 50% (IC50) or 90% (IC90) as determined by MTT assay after 120 h of incubation. The sublines display significantly increased YM155 resistance. Numerical values are presented in Appendix A. * *p* < 0.05 relative to UKF-NB-3.

**Figure 2 cancers-12-01080-f002:**
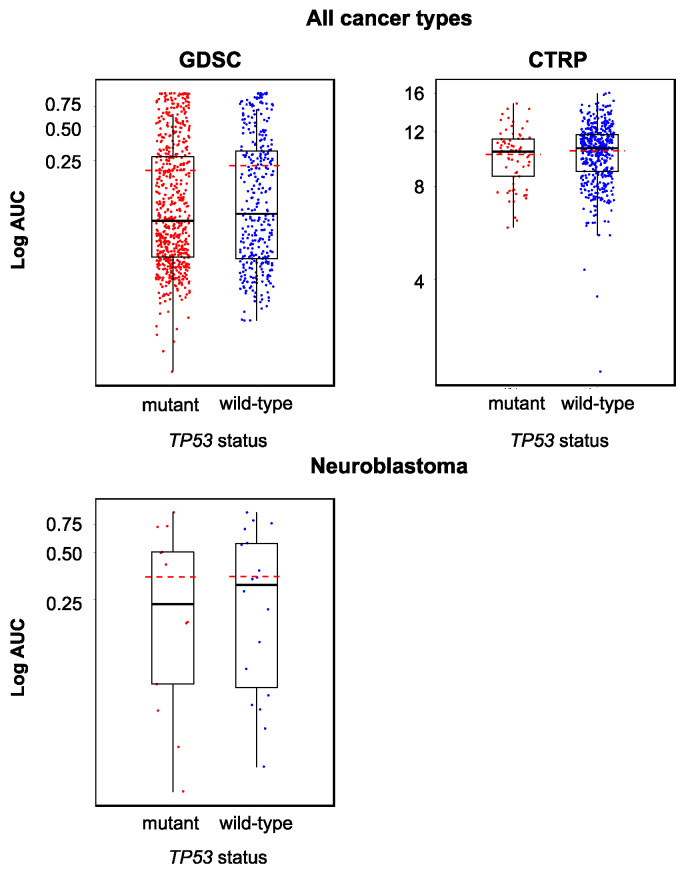
YM155 sensitivity in *TP53* wild-type and *TP53* mutant cancer cell lines based on the analysis of GDSC and CTRP data, both determined across all investigated cancer types/cell lines (GDSC, *p* = 0.458; CTRP, *p* = 0.216) and in a neuroblastoma-specific analysis (GDSC, *p* = 0.922; all 12 neuroblastoma cell lines in the CTRP harbor wild-type *TP53*). The comparisons did not reveal significant differences.

**Figure 3 cancers-12-01080-f003:**
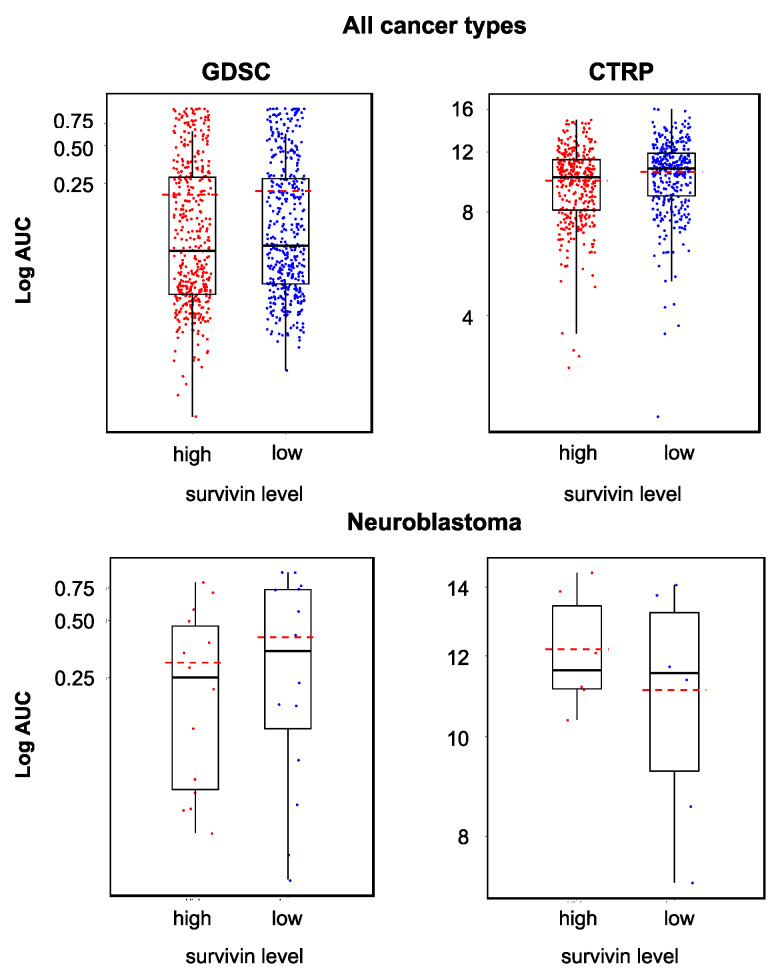
YM155 sensitivity in cell lines characterized by high or low survivin expression based on the analysis of GDSC and CTRP data, both determined across all investigated cancer types/cell lines (GDSC, *p* = 0.048; CTRP, *p* < 0.001) and in a neuroblastoma-specific analysis (GDSC, *p* = 0.425; CTRP, *p* = 0.699). Across all cell lines, high survivin expression was associated with increased YM155 sensitivity, but the YM155 sensitivity of individual cell lines could not be predicted based on the survivin levels.

**Figure 4 cancers-12-01080-f004:**
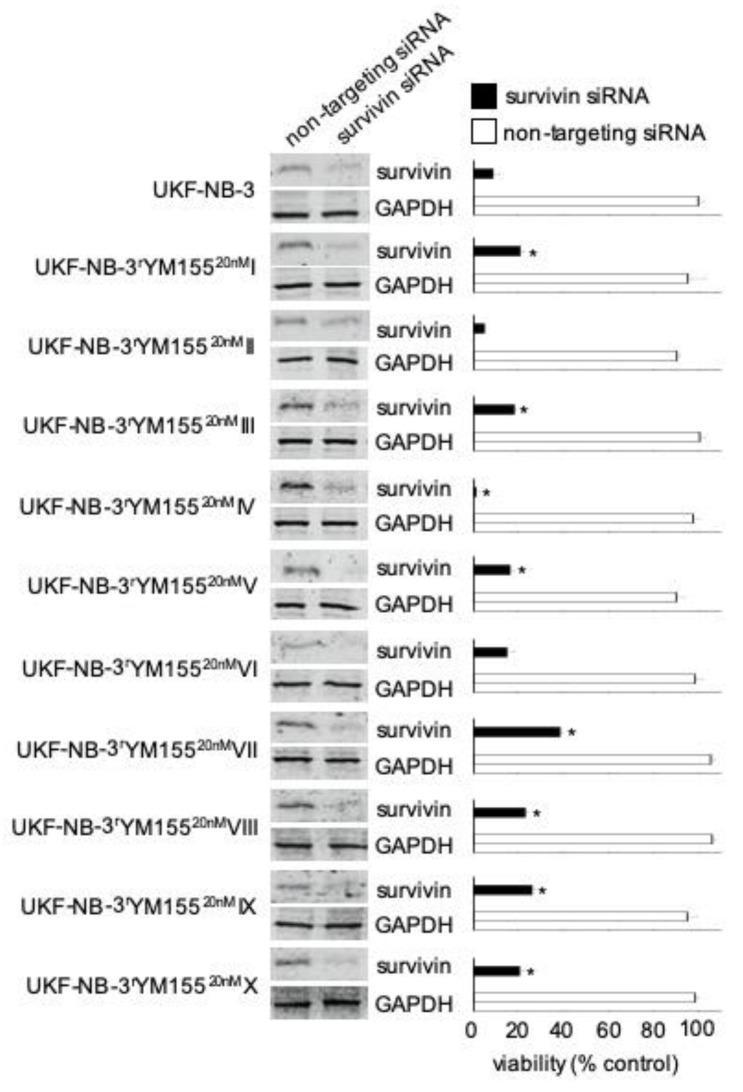
Effects of siRNA-mediated BIRC5/survivin depletion on the viability of UKF-NB-3 and its YM155-adapted sublines. Western blots confirm reduced survivin levels 48 h post-transfection. Viability of cells transduced with siRNA directed against BIRC5/survivin or non-targeting siRNA was determined relative to untreated control cells 168 h post transfection by MTT assay. The cell lines displayed varying levels of sensitivity to survivin depletion. * *p* < 0.05 relative to untreated cells. Uncropped blots are shown in Appendix A.

**Figure 5 cancers-12-01080-f005:**
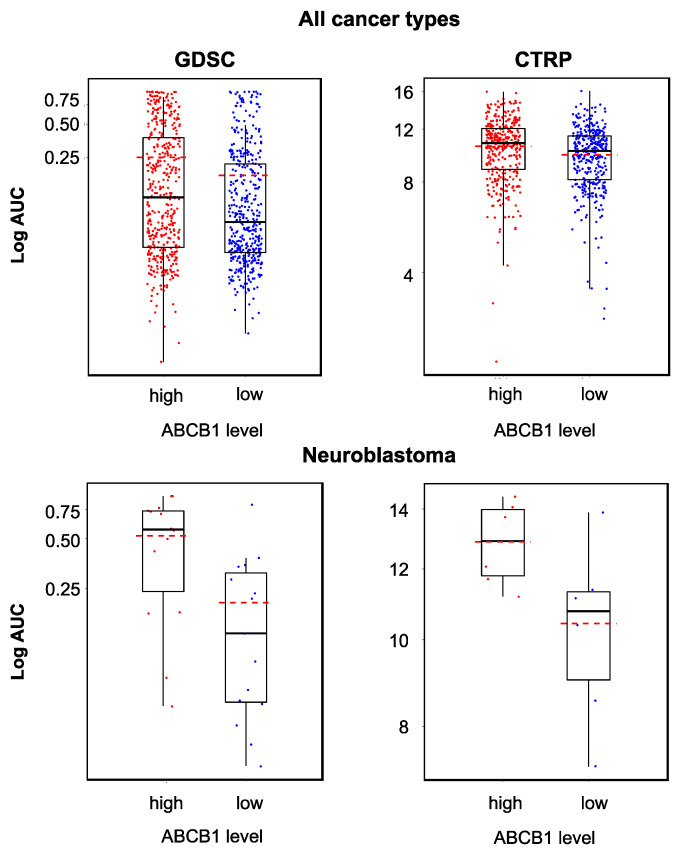
YM155 sensitivity in cancer cell lines characterized by high or low ABCB1 expression based on the analysis of GDSC and CTRP data, both determined across all investigated cancer types/cell lines (GDSC, *p* < 0.001; CTRP, *p* < 0.001) and in a neuroblastoma-specific analysis (GDSC, *p* = 0.006; CTRP, *p* = 0.04). High ABCB1 expression was associated with decreased YM155 sensitivity, but the YM155 sensitivity of individual cell lines could not be predicted based on the ABCB1 levels.

**Figure 6 cancers-12-01080-f006:**
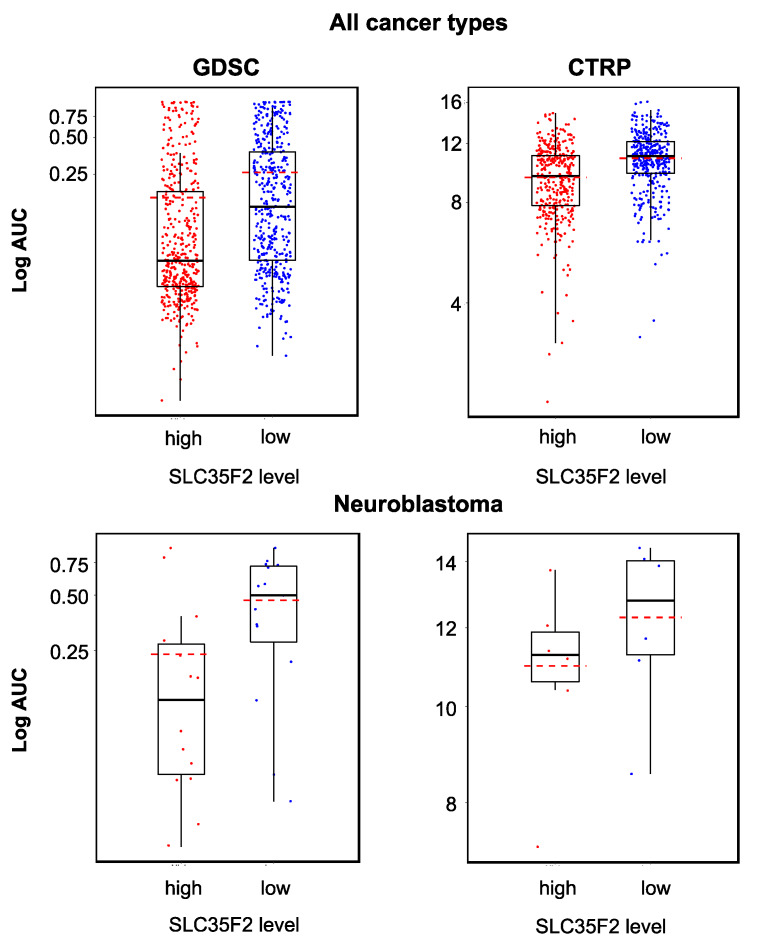
YM155 sensitivity in cancer cell lines characterized by high or low SLC35F2 expression based on the analysis of GDSC and CTRP data, both determined across all investigated cancer types/cell lines (GDSC, *p* < 0.001; CTRP, *p* < 0.001) and in a neuroblastoma-specific analysis (GDSC, *p* = 0.033; CTRP, *p* = 0.310). High SLC35F2 expression was associated with increased YM155 sensitivity, but the YM155 sensitivity of individual cell lines could not be predicted based on their SLC35F2 levels.

**Figure 7 cancers-12-01080-f007:**
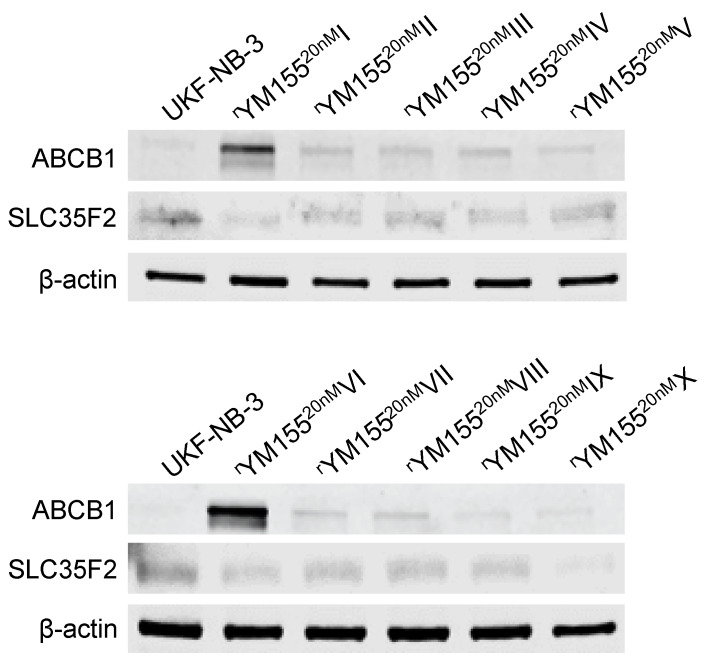
Representative Western blots indicating cellular levels of ABCB1 and SLC35F2 in UKF-NB-3 and YM155-adapted UKF-NB-3 sublines. YM155-adapted sublines are typically characterized by increased ABCB1 levels and decreased SLC35F2 levels. Uncropped blots are shown in Appendix A.

**Figure 8 cancers-12-01080-f008:**
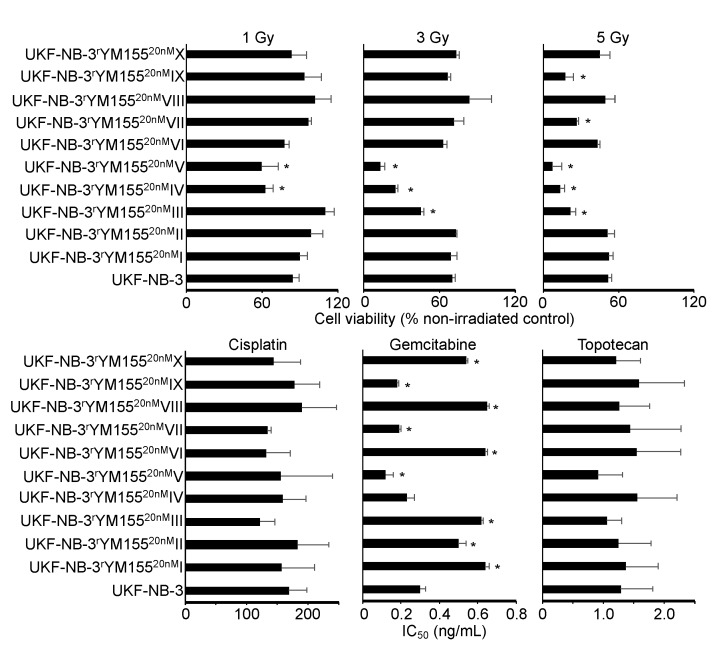
Sensitivity of UKF-NB-3 and its YM155-adapted sublines to irradiation and DNA-damaging drugs. The radiation response was determined 72 h after irradiation with 1, 3, or 5Gy by MTT assay. Drug concentrations that reduce cell viability by 50% (IC_50_) were determined by MTT assay after 120 h of incubation. The YM155-adapted sublines did not display increased resistance to DNA damage induced by radiation or the drugs cisplatin, gemcitabine, or topotecan. * *p* < 0.05 relative to UKF-NB-3.

**Figure 9 cancers-12-01080-f009:**
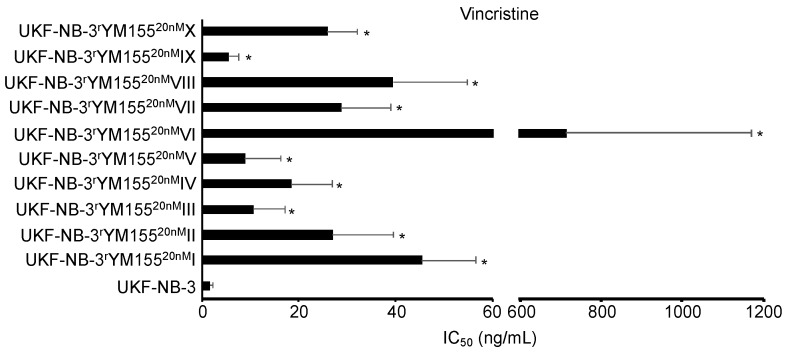
Vincristine concentrations that reduce cell viability by 50% (IC_50_) were determined by MTT assay after 120 h of incubation. YM155-adapted sublines consistently displayed decreased vincristine sensitivity relative to UKF-NB-3. The vincristine IC_50_ values varied considerably. * *p* < 0.05 relative to UKF-NB-3.

**Table 1 cancers-12-01080-t001:** Top 10 genes whose expression is most strongly correlated with the YM155 AUC in the Genomics of Drug Sensitivity in Cancer (GDSC) database and the Cancer Therapeutics Response Portal (CTRP) as indicated by the Pearson correlation coefficient.

GDSC			CTRP		
Gene	Correlation Coefficient	FDR ^1^	Gene	Correlation Coefficient	FDR
ABCB1	0.3792069	2.82 × 10^−6^	ABCB1	0.3624858	2.70 × 10^−6^
FABP1	0.2972293	5.64 × 10^−6^	CST3	0.3603422	5.39 × 10^−6^
CDX2	0.2927057	8.46 × 10^−6^	AKR1C3	0.3503176	8.09 × 10^−6^
DDC	0.2922995	1.13 × 10^−5^	EPS8	0.3453892	1.08 × 10^−5^
CDH17	0.2652645	1.41 × 10^−5^	ABHD2	0.3309384	1.35 × 10^−5^
ANKS4B	0.2637047	1.97 × 10^−5^	S100A6	0.3229691	1.62 × 10^−5^
MYO1A	0.2626518	2.26 × 10^−5^	ATP1B1	0.3111308	1.89 × 10^−5^
PHGR1	0.2609317	2.54 × 10^−5^	CD63	0.310712	2.16 × 10^−5^
A1CF	0.252912	2.82 × 10^−5^	AKR1C1	0.3073306	2.43 × 10^−5^
GUCY2C	0.2513048	3.10 × 10^−5^	ACVR1	0.30124	9.44 × 10^−5^

^1^ false discovery rate.

**Table 2 cancers-12-01080-t002:** Top 10 genes whose expression is most strongly inversely correlated with the YM155 AUC in the Genomics of Drug Sensitivity in Cancer (GDSC) database and the Cancer Therapeutics Response Portal (CTRP) as indicated by the Pearson correlation coefficient.

GDSC			CTRP		
Gene	Correlation Coefficient	FDR ^1^	Gene	Correlation Coefficient	FDR
SLC35F2	−0.2643809	1.69 × 10^−5^	SLC35F2	−0.3868041	9.17 × 10^−5^
ALKBH8	−0.2042557	0.000121	CD19	−0.349156	8.90 × 10^−5^
CWF19L2	−0.1926243	0.000180	CD79B	−0.346035	8.63 × 10^−5^
RCSD1	−0.1855956	0.000226	SPIB	−0.341887	8.36 × 10^−5^
P2RY8	−0.1834937	0.000257	SNX22	−0.338893	8.09 × 10^−5^
RGS19	−0.1831031	0.000259	TCL1A	−0.338393	7.82 × 10^−5^
FLI1	−0.1825892	0.000268	LOC100130458	−0.337940	7.55 × 10^−5^
VAV1	−0.1819569	0.000273	BLK	−0.330471	7.28 × 10^−5^
ATM	−0.1816322	0.000276	CD79A	−0.324465	7.01 × 10^−5^
ARHGAP19	−0.1813872	0.000282	VPREB3	−0.322878	6.74 × 10^−5^

^1^ false discovery rate.

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
