# Peer review of "YM155-Adapted Cancer Cell Lines Reveal Drug-Induced Heterogeneity and Enable the Identification of Biomarker Candidates for the Acquired Resistance Setting"

_cancers, 2020, doi:10.3390/cancers12051080_

Round 1

Reviewer 1 Report

I understand this study took quite an amount of effort to create 10 sublines (As the authors mention), yet there is no complete survivine-resistance, thus somehow ended up unfortunately with negative results. Yet, some heterogeneity in the survivine-dependence is existed (the authors claim it is important), but they did not clarify the basis for the heterogeneity further. That seems to mean the double negativeness of the study.

Although cellular ABCB1 and SLC35F2 levels do not enable the reliable prediction of the YM155 sensitivity status, an increase in ABCB1 and/ or a decrease in SLC35F2 in response to YM155 treatment are likely to indicate resistance formation

If so, they should test at least this possibility by knocking down these genes.

It would be more relevant if they could validate those resistance clones in vivo.

Author Response

The authors do not understand, why the reviewer would suggest that our study would report negative results. The finding that YM155-resistant cells retain sensitivity to survivin depletion is a novel finding, and we do not understand how you might regard this as a negative result. Along similar lines, we agree that it would be nice to know the mechanisms underlying the heterogeneity of the sublines, but this knowledge gap does not make the discovery of this heterogeneity a negative result. The authors also feel that there is likely not one mechanism responsible for this heterogeneity, which could be reasonably deciphered in the scope of this study. Finally, even though we do not feel that we would have reported negative findings here, we feel that negative findings can be extremely valuable, which is a notion that is shared by an increasing number of scientists and eminent research journals.

The effects of ABCB1 and SLC35F2 on YM155 efficacy have been previously shown by us and others by a range of methods including depletion experiments (e.g. Eur J Cancer. 2012 Mar;48(5):763-71; Nat Chem Biol. 2014 Sep;10(9):768-773; Cell Death Dis. 2016 Oct 13;7(10):e2410; Sci Rep. 2017 Jun 8;7(1):3091). Hence, we do not feel that such experiments would further increase the impact or plausibility of our study.

Reviewer 2 Report

The revised version of the manuscript appears to fulfill previous comments.

Author Response

Thank you.

Reviewer 3 Report

All the comments have been addressed and manuscript improved from the previous version.

Author Response

Thank you.

Reviewer 4 Report

Review of the manuscript “YM155-adapted cancer cell lines reveal drug-induced heterogeneity and enable the identification of biomarker candidates for the acquired resistance setting” by Martin Michaelis and coauthors submitted to “Cancer”, MDPI

The formation of acquired resistance is a key question for the selection of a strategy for a systemic drug therapy in metastasized cancer diseases. The authors investigated whether the use of the panel of YM155-86 adapted cell lines may show that the cellular survivin levels and the TP53 status are indicators of the resistance formation. Recent data indicate that the suppression of survivin expression is an important mechanism by which YM155 exerts its anticancer effects. However, further research is needed to have a deeper insight into the problem.  This is an important area of investigation in cancer research and the results will be interesting for the readership of “Cancer”. 

The following corrections should be done:

Introduction:

Lines 79-81. “To see whether we can obtain information from our acquired resistance models that cannot be identified from traditional approaches using non-adapted cell lines, we also analysed YM155 response data from the two large pharmacogenomics screens Genomics of Drug Sensitivity in Cancer (GDSC) and Cancer Therapeutic Response Portal (CTRP) [29,30]”

This is hard to read sentence which should be cut in 2-3 shorter sentence to make it easier to readers.

Results.

Figure 2. “Only p53 wild type neuroblastoma cells lines were analyzed in the CTRP”. This sentence written in large font is not located in a right place on Figure 2. It should be moved to the figure legend, text or elsewhere.  

Line 119 :”The cellular p53 levels did also not differ consistently…” should ,be corrected as “The cellular p53 levels also did not differ consistently…”

Figure 4. The writing on Figure 4 should be more clear

Lines 217-218:” These data do not suggest a dominant role of DNA damage induction in the course of the anti-neuroblastoma activity of YM155 in UKF-NB-3 cells”.It is not clear what the authors want to say by “in the course”? May be “associated”?

Discussion. “Moreover the use of multiple sublines provided a pioneering glimpse onto the remarkable heterogeneity of the resistance formation process, even though all resistant sublines were derived from the same parental cell line”.

The authors are too pathetic in this statement. It would be better to replace “pioneering” by “novel” and “remarkable” by significant.

Methods Lines 310-311: “Cells were routinely tested for mycoplasma contamination” The authors should give a method used for mycoplasma contamination test.

4.4 Western blot. What secondary antibodies were used? Dilution?

Author Response

Review of the manuscript “YM155-adapted cancer cell lines reveal drug-induced heterogeneity and enable the identification of biomarker candidates for the acquired resistance setting” by Martin Michaelis and coauthors submitted to “Cancer”, MDPI

The formation of acquired resistance is a key question for the selection of a strategy for a systemic drug therapy in metastasized cancer diseases. The authors investigated whether the use of the panel of YM155-86 adapted cell lines may show that the cellular survivin levels and the TP53 status are indicators of the resistance formation. Recent data indicate that the suppression of survivin expression is an important mechanism by which YM155 exerts its anticancer effects. However, further research is needed to have a deeper insight into the problem.  This is an important area of investigation in cancer research and the results will be interesting for the readership of “Cancer”.

The following corrections should be done:

Introduction:

Lines 79-81. “To see whether we can obtain information from our acquired resistance models that cannot be identified from traditional approaches using non-adapted cell lines, we also analysed YM155 response data from the two large pharmacogenomics screens Genomics of Drug Sensitivity in Cancer (GDSC) and Cancer Therapeutic Response Portal (CTRP) [29,30]”

This is hard to read sentence which should be cut in 2-3 shorter sentence to make it easier to readers.

Authors’ response:

This was done. The passage reads now (p. 2, line 79-84):

"Moreover, acquired resistance models may provide information that cannot be gained from the comparison of non-adapted cell lines with varying resistance status. To investigate whether this is the case, the findings from the YM155-adapted UKF-NB-3 sublines were compared to data from the two large pharmacogenomics screens Genomics of Drug Sensitivity in Cancer (GDSC) and Cancer Therapeutic Response Portal (CTRP), which use non-adapted cancer cell lines [29,30]."

Results.

Figure 2. “Only p53 wild type neuroblastoma cells lines were analyzed in the CTRP”. This sentence written in large font is not located in a right place on Figure 2. It should be moved to the figure legend, text or elsewhere. 

Authors’ response:

This was done. The Figure legend reads know (p. 4, lines 115-119):

"Figure 2. YM155 sensitivity in p53 wild-type and p53 mutant cancer cell lines based on the analysis of GDSC and CTRP data, both determined across all investigated cancer types/ cell lines (GDSC, p = 0.458; CTRP, p = 0.216) and in a neuroblastoma-specific analysis (GDSC, p = 0.922; all 12 neuroblastoma cell lines in the CTRP harbour wild-type TP53). The comparisons did not reveal significant differences."

Line 119 :”The cellular p53 levels did also not differ consistently…” should ,be corrected as “The cellular p53 levels also did not differ consistently…”

Authors’ response:

This was done (now p. 5, line 126).

Figure 4. The writing on Figure 4 should be more clear

Authors’ response:

This was done. The paragraph reads now (p. 7, line 153-160):

"Our previous findings had suggested that YM155 predominantly exerts its anti-neuroblastoma effects via suppression of survivin expression [16]. Seven of the ten YM155-adapted UKF-NB-3 sublines (I, III, V, VII, VIII, IX, X) displayed decreased sensitivity to siRNA-mediated survivin depletion, indicating that they have developed on-target resistance. However, two sublines were similarly sensitive as parental UKF-NB-3 cells (II, VI), and one subline (IV) was more sensitive to survivin depletion (Figure 4, Figure S4). This shows that the YM155 resistance mechanisms differ between the individual UKF-NB-3 sublines. Notably, the viability of all sublines is still affected by survivin depletion, which shows that they have retained some level of survivin dependence."

Lines 217-218:” These data do not suggest a dominant role of DNA damage induction in the course of the anti-neuroblastoma activity of YM155 in UKF-NB-3 cells”.It is not clear what the authors want to say by “in the course”? May be “associated”?

Authors’ response:

The sentence was rephrased to increase its clarity (p. 13, lines 232-234):

"These data do not suggest that the activity of YM155 against UKF-NB-3 cells would predominantly depend on DNA damage induction."

Discussion. “Moreover the use of multiple sublines provided a pioneering glimpse onto the remarkable heterogeneity of the resistance formation process, even though all resistant sublines were derived from the same parental cell line”.

The authors are too pathetic in this statement. It would be better to replace “pioneering” by “novel” and “remarkable” by significant.

Authors’ response:

This was done (now line 311).

Methods Lines 310-311: “Cells were routinely tested for mycoplasma contamination” The authors should give a method used for mycoplasma contamination test.

Authors’ response:

The information was added (now lines 328/329).

4.4 Western blot. What secondary antibodies were used? Dilution?

Authors’ response:

The information on the secondary antibodies was added (now lines 377-387).

Round 2

Reviewer 1 Report

The process of isolating resistant clones is time-consuming with simple tissue culture procedure, but the process for characterizing the clones requires more efforts to find new insights on a drug resistance.  I understand this study took quite an amount of time to create 10 sublines, yet there is no complete survivine-resistance, thus somehow ended up unfortunately with negative results. Yet, some heterogeneity in the survivine-dependence is existed (the authors claim it is important), but they did not clarify the molecular basis for the heterogeneity further. The study, therefore, unfortunately, provides only a weak and unclear conclusion.

In the authors' replay, they mention as,

Although cellular ABCB1 and SLC35F2 levels do not enable the reliable prediction of the YM155 sensitivity status, an increase in ABCB1 and/ or a decrease in SLC35F2 in response to YM155 treatment are likely to indicate resistance formation

If so, they should test at least this possibility by knocking down these genes.

This manuscript is a resubmission of an earlier submission. The following is a list of the peer review reports and author responses from that submission.

Round 1

Reviewer 1 Report

In this manuscript as the extension of their previous study, the authors established 10 additional sublines of a neuroblastoma cell line, UKF-NB-3, based on the adaptation to YM155, a potent anti-cancer drug discovered as a survivin suppressor. These 10 sublines were subjected to study for the characteristics known to be associated with resistance to the drug, including p53 status, sensitivity to survivin-depletion, ABCB1 and SLC35F2 expression levels, and the response to the DNA damage-inducing reagents, that revealed variations and heterogeneity. Additionally the two pharmacogenomic databases, CTD2 and GDSC were used to analyze genetic and transcriptomic correlation with drug sensitivity for p53 mutation and expression of survivin, ABCB1 and SLC35F2, that confirmed their relevance to the drug response.

The weakness of this study is the lack of novelty impact in the resistance formation mechanism and the resistance prediction to YM155 and the translational relevance as a preclinical study. While they describe the variations and thus heterogeneity in the development of acquired adaptations to the drug for factors known to associate with sensitivity, there is no clear biomarker prediction establishment, and the relevance is not certain as it could be only in vitro and/or only in this one cell line in this limited experiment subject. In fact, as YM155 has been already under clinical trials, it would be important to make efforts on in vivo validations with xenograft models using human cell lines or PDX.

Here shows some more comments.

Figure 4 all sublines are still impaired of their viability upon survivin depletion, maybe exhibit cyto-static arrest instead of cytotoxic response, but not fully resistant to the survivin depletion and they still need survivin for a full growth as compared to the GAPDH, suggesting that the survivin is really essential and indispensable for the cell viability and thus replacement by another dependent may not be easily happen in this cell line.

Si-RNA effect should be test for another sequence to validate the specificity.

Figure 1 They should show the survivin expression upon YM155 treatment together with other IPAs including Mcl-1 for all sublines. The effects of the drug on survivin depletion may associate with the change of ABCB1 and SLC35F2 as expected from previous studies.

The authors should consider and at least mention potential of mTORC1 to be involved in the drug sensitivity as the recent study reported the mTORC regulation by YM155.

Early Cellular Responses of Prostate Carcinoma Cells to Sepantronium Bromide (YM155) Involve Suppression of mTORC1 by AMPK

David Danielpour, Zhaofeng Gao, Patrick M. Zmina, Eswar Shankar, Benjamin C. Shultes, Raul Jobava, Scott M. Welford & Maria Hatzoglou

Scientific Reports volume 9, Article number: 11541 (2019)

Reviewer 2 Report

In this manuscript, the authors report their study on the resistance mechanisms of a neuroblastoma cell line to an anti-cancer drug candidate YM155, a small heterocyclic molecule. YM155 is presented as a surviving inhibitor potentially active against several cancers, with several papers dedicated to its activity against neuroblastoma cells.

The work presented in this paper is focused on the possible identification of biomarkers to assess resistance formation in acquired resistance setting as well as to predict such outcome.

For such studies, the authors have developed a series of YM155 -adapted UKF-NB-3 neuroblastoma sublines. They report their findings on several cellular biomarkers such as ABCB1, SLC35F2, TP53 or survivin. Sublines also exhibit large heterogeneity of responses, particularly to a variety of other anti-cancer drugs.

The rational and outcome of the study appear to be well explained and discussed. Several WB results presented as supplementary materials are nevertheless of quite weak quality.

Studies on resistance mechanisms to anti-cancer drugs are numerous and papers on such studies exhibit a quite large variety of outcome. The title of the manuscript should be more precise since all results presented here are only dedicated to YM155 resistance formation.

It might be changed as “YM155-adapted cancer cell lines reveal…..”.

Reviewer 3 Report

The manuscript showed the heterogenicity in drug adapted cancel cells. However, there are few minor comments required to be corrected:

Page 1; Line 23-24: Sentence should be rephrased. "Survivin is a drug target and its suppressant YM155, a drug candidate mainly used for high-risk neuroblastoma".

Page 1; Line 25-26: Is there any significant % increase or decrease?

Page 2; Line 45: It should be better to use full name at first "Sepantronium bromide (YM155)"

Page 3; Line 101: Need to explain the figure outcome

Page 4; Line 111: Results in the figure should be described rather than just writing the objective.

Page 16; Line 297: What was passage of cells?

Page 16; Line 306: The cells viability method is too short and need to explain in more brief even adapted from previous method.

Page 17; Line 352: Sentence shouldn't be started with numeric values

Page 18; Line 384: Two-way ANOVA?

Reviewer 4 Report

Review of a manuscript “Drug-adapted cancer cell lines reveal drug-induced heterogeneity and enable the identification of biomarker candidates for the acquired resistance setting” by Martin Michaelis submitted to “Cancers”, MDPI.

The authors address the complexity of the drug resistance formation process for neuroblastoma. They investigated survivin as a drug target and the survivin suppressant YM155 as a drug candidate for high-risk neuroblastoma. For this purpose they established and characterized YM155-adapted UKF79 NB-3 neuroblastoma cell lines. The authors also analyzed YM155 response data from the large pharmacogenomics screens, GDSC and CTRP. The topic of their investigation is very important, and the results presented in the manuscript will be interesting for the readers of “Cancers”, MDPI.

The following corrections should be done.

Keywords: the authors should add BIRC5 gene

Results:

Line 92: “Representative photos of the morphology of the project cell lines are presented in Figure S1 and the doubling times in Table S1.”

The authors should begin the Result section with the reference to Figures placed in the main text, but not to the data from the Supplementary Materials.

Lines 94-99. This part of the result section is overloaded with figures and abbreviations and therefore is hard to read. The authors should rewrite this part to make it more “reader-friendly”. The stress should be on main, most essential findings.

Lines 150-151:”This shows that a majority of the YM155-resistant cell lines have developed on-target resistance. It also indicates that the YM155 resistance mechanisms differ between the individual UKF-NB-3 sublines”.

The authors begin two sentences by “this” and “it”. They should express more exactly what they mean, be replacing “this” and “it” on more clear terms.

Methods  

Line 204 “Doubling times were then calculated using http://www.doubling-time.com/compute.php”. The authors should explain the principle of the method, but not just refer the reader to a website.

Line 333. 4.4 Western blot

The authors should give the dilution of antibodies used for Western blot.

Lines 385-386

“Three or more groups were compared by ANOVA followed by the Student-Newman-Keuls test. P values lower than 0.05 were considered to be significant”.

The authors should clarify what they mean by saying “Three or more groups”

Lines 434 -435  

“In the text, reference numbers should be placed in square brackets [ ], and placed before the punctuation; for example [1], [1–3] or [1,3]. For embedded citations in the text with pagination, use both parentheses and brackets to indicate the reference number and page numbers; for example [5] (p. 10), or [6] (pp. 101–105).”

It looks like the authors transferred the pieces of Instruction for authors in their manuscript. We doubt that these detailed instructions will be interesting for readers.